# PoreVision: A Program for Enhancing Efficiency and Accuracy in SEM Pore Analyses of Gels and Other Porous Materials

**DOI:** 10.3390/gels11020132

**Published:** 2025-02-13

**Authors:** Levi M. Olevsky, Mason G. Jacques, Katherine R. Hixon

**Affiliations:** 1Thayer School of Engineering, Dartmouth College, Hanover, NH 03755, USA; levi.olevsky.th@dartmouth.edu; 2College of Engineering and Physical Sciences, University of New Hampshire, Durham, NH 03824, USA; mason.g.jacques@dartmouth.edu; 3Geisel School of Medicine, Dartmouth College, Hanover, NH 03755, USA

**Keywords:** pore size, pore analysis, scaffold, Python, tissue engineering

## Abstract

Porous gels are frequently utilized as cell scaffolds in tissue engineering. Previous studies have highlighted the significance of scaffold pore size and pore orientation in influencing cell migration and differentiation. Moreover, there exists a considerable body of research focused on optimizing pore characteristics to enhance scaffold performance. However, current methods for numerical pore characterization typically involve expensive machines or manual size measurements using image manipulation software. In this project, our objective is to develop a user-friendly, versatile, and freely accessible software tool using Python scripting. This tool aims to streamline and objectify pore characterization, thereby accelerating research efforts and providing a standardized framework for researchers working with porous gels. Our group found that first-time users of PoreVision and ImageJ take similar amounts of time to use both programs; however, PoreVision is capable of handling larger datasets with reduced variability. Further, PoreVision users exhibited lower variability in area and orientation measurements compared to ImageJ, while perimeter variability was similar between the two. PoreVision showed higher variability in average measurements, likely due to its larger sample size and broader range of pore sizes, which may be missed in ImageJ’s manual scanning approach. By facilitating quantitative analysis of pore size, shape, and orientation, our software tool will contribute to a more comprehensive understanding of scaffold properties and their impact on cellular behavior. Ultimately, we aim to aid researchers in the field of tissue engineering with a user-friendly tool that enhances the reproducibility and reliability of pore characterization analyses.

## 1. Introduction

Tissue engineering is a multidisciplinary field that combines cells, growth factors, and scaffolds to develop biological substitutes for defective tissue [1]. Scaffolds are biomaterials that serve as supportive frameworks, providing essential sites for cell attachment and tissue growth [2]. Scaffolds play critical roles in tissue development, including architectural support, structural reinforcement, bioactive signaling, growth factor delivery, and physical space for tissue remodeling [3,4,5,6]. The architecture of scaffolds is particularly important, influencing cell behavior, adhesion, and proliferation, as well as facilitating vascularization and nutrient exchange [7,8,9,10]. Accordingly, many gel scaffolds feature porous structures whose shape and size directly impact cell viability and tissue growth [11,12,13,14,15]. Thus, the ability to quantitatively assess pore morphology is essential for understanding its correlation with cellular behavior and tissue viability.

Numerous techniques for quantifying pore morphology have been developed. The gravimetric method offers a straightforward approach to assess scaffold porosity, providing insight into the amount of empty space within a scaffold [16]. However, its limitations include a lack of precision in volume measurement and an inability to characterize the actual size distribution of pores, leaving ambiguity in the prevalence of small versus large pores [17]. Mercury porosimetry is an alternative technique, yielding measurements such as total pore volume fraction, average pore diameter, and pore size distribution [17,18]. This method avoids user bias but may pose challenges for scaffolds with delicate or thin pores, potentially compromising accuracy [18]. Moreover, the use of mercury raises concerns due to its toxicity and associated costs [17,18]. Other methodologies, such as capillary flow porometry and permeability-based methods, involve fluid dynamics to calculate pore diameter [17,19,20]. While these techniques offer advantages, they often assume a simplified pore geometry, such as a cylindrical shape with a constant radius, which may not accurately represent true pore structures [17,19,20].

In addition to the methods mentioned, several computer-aided and mathematical approaches have been developed for quantifying pore morphology in scaffolds. One common method involves sectioning samples and utilizing specialized software for analysis [21,22,23,24,25]. This approach enables precise measurement of pore dimensions and shape using scanning electron microscopy (SEM) images, but it is prone to user bias and lacks a standardized definition of “pore diameter” [21,22,23,24,25]. Moreover, manual measurements may suffer from limited sample sizes, potentially affecting the reliability of results [26,27]. Some microscopy systems are equipped with software capable of automatically measuring pore sizes, offering a more efficient alternative to manual methods, but these programs are limited to the specific microscopy systems [28,29]. Additionally, ImageJ, a widely used image analysis software, can be employed for manual or automatic pore size measurements for little to no cost [30]. However, certain workflows and add-ons may require expertise and may not be user-friendly for all researchers [30,31,32,33,34]. Measurement workflows that are native to ImageJ include using selections (oval, polygon, straight, etc.) to outline and measure different pore characteristics (area, perimeter, diameter) as well as using the “analyze particles” feature to quantify pore measurements in 8-bit, binary images [35,36]. Other methods use ImageJ plugins that may not be user-friendly, such as deep image processing or granulometry [31,33]. Finally, micro-computed tomography (microCT) represents another powerful tool for pore characterization, offering three-dimensional visualization of scaffold structures [37,38,39]; however, its adoption may be hindered by its high cost [40].

Computational and mathematical models present a cost-effective solution that could benefit research groups studying porous gel scaffolds. Despite their use and availability, many current methodologies lack standardized measurements or fail to detail the specific approach used for pore size analysis [30,41,42]. To address this gap, there is a need for the development of a user-friendly software tool that provides standardized analysis of pore morphology, thereby enhancing the reproducibility and objectivity of research outcomes. In this paper, our objective is to introduce a freely accessible software tool that aims to streamline and standardize pore characterization processes, offering an intuitive interface that facilitates efficient analysis of pore morphology.

PoreVision (version 1) was developed in response to the inherent challenges of manually characterizing porous gels from SEM images, a process known to be time-consuming and prone to inconsistencies [43,44]. Even when two researchers follow the same procedure, significant discrepancies in pore measurements can arise, highlighting the need for a more efficient and reliable solution. PoreVision addresses this need by enabling researchers to characterize pores easily and consistently, facilitating accurate and high-throughput analysis. PoreVision is written entirely in Python, ensuring ease of modification by interested parties. Moreover, it is constructed entirely from free software, making this powerful analysis tool accessible to researchers operating on limited budgets.

PoreVision incorporates shape and orientation analysis capabilities that can provide insights into critical pore characteristics that may influence scaffold performance, including cell viability, differentiation, and migration [42,45,46,47,48]. Our aim is to aid researchers with an intuitive tool that unlocks valuable information from scaffold research, ultimately accelerating the pace of discovery in this field.

## 2. Results and Discussion

PoreVision was developed to tackle the challenges of manually characterizing porous gels from SEM images, a process prone to time-consuming errors and inconsistencies [43,44]. Even with standardized procedures, significant disparities in pore measurements can occur, highlighting the need for a more efficient solution. PoreVision offers researchers a user-friendly and consistent approach to pore characterization, facilitating precise and high-throughput analysis. Written entirely in Python, it allows for easy modification and is accessible to researchers on limited budgets [49,50,51]. Additionally, PoreVision includes shape and orientation analysis capabilities, providing insights into critical pore characteristics that affect scaffold performance, such as cell viability, differentiation, and migration [42,45,46,47]. Our goal is to provide researchers with an intuitive tool that accelerates discovery in scaffold research.

### 2.1. How to Use the Program

PoreVision was designed to be as intuitive as possible, but for maximum clarity, a full walkthrough of features is provided here. The program can be found in the Appendix A. After downloading the ZIP file, the program can be found in the ‘PoreVision Program’ folder, titled PoreVision. When starting the program, the user encounters two buttons, ‘Load Image’, and ‘Information’. ‘Information’ provides information about the program and links to this paper. When an image (PNG or TIFF file) is selected through a file dialog, it is displayed in the window along with a box on the right side of the screen, called “Analysis Settings” (Figure 1). Each analysis setting is explained in detail in Table 1.

An SEM image will typically have a region on one side that contains the scalebar and other information. If the image is from a Tescan SEM, try “Off”. If using a different SEM, the user should crop this section out prior to loading into PoreVision and use the “Supply Scale” option “On” or “Pixel” instead. “On” allows the user to supply PoreVision with the scalebar length (in pixels) and reading (in micrometers). ImageJ or MS Paint can be used to determine scalebar length. “Pixel” will forgo any unit conversions and perform all analysis in terms of pixels.

Pressing the green ‘Analyze’ button will display the detected pores in the window, and create another box labeled “Display Settings” (Figure 2) below the first with the following adjustable settings (Table 2):

Changing any of these options will automatically refresh the image. A toggle button ‘View’ also allows the user to quickly swap between the analyzed and original images. The button to the left of ‘View’ allows the user to change what image is shown “underneath” when ‘View’ is pressed. For example, if ‘Original’ is selected, the underlying image will be the originally supplied image when ‘View’ is pressed. If ’CLAHE’ is selected, the processed image using the CLAHE algorithm will be displayed when ’View’ is pressed. See Section 2.4 for explanations and example images of each type of image processing.

A third box will also appear below the first two, labeled “Results” (Figure 3). This box displays the analysis results for that analysis pass, consisting of the number of detected pores, the smallest pore, the largest pore, the mean pore area, the standard deviation of pore area, the mean perimeter, the mean IPR, and the mean pore density (unless in pixel mode). The radio button at the top of the box will allow the user to display results for that analysis pass or for the saved contours. Note that the mean feret diameter is not shown in the “Results” section but can be found in the exported csv. Other statistics of interest can be analyzed using the individual pore data in the CSV file.

Clicking on a contour creates another box that relates some information about that specific contour, consisting of center, area, perimeter, IPR, and orientation (Figure 4). The user is also presented with buttons to remove or save contours. The ‘Remove’ button allows the user to remove a single contour, whereas the ‘[range]’ button adjacent to ‘Remove’ allows the user to remove a range of contours. The same feature is replicated for saving contours.

Saving a contour will remove the pore from that analysis pass and place it into the “Saved” list. To inspect saved pores, change the radio button in the results box to “Saved.” Clicking on a saved contour will also display the analysis pass used to produce that contour, similar to how they would appear in an exported CSV file.

Clicking on the pink ‘Save PNG’ button at the bottom of the window will save a PNG file of whichever image is currently displayed on the screen. Clicking the pink ‘Save CSV’ button will create and export a CSV file of the list of contours currently selected by the Results radio button (detected or saved).

### 2.2. Validation

#### 2.2.1. Sample Workflow

Below, we present a sample workflow explaining the step-by-step process of utilizing PoreVision to analyze a porous gel SEM image.

Begin by opening the program. Click ‘Choose File’ and select the image you would like to analyze (Figure 5). The sample image is included in Appendix A.

To set the scale, navigate to the ‘Supply Scale’ section and choose the appropriate option. For the purpose of this sample workflow, we will select the ‘On’ option. A window will then appear prompting you to input the scale reading and scale length (Figure 6). It is worth noting that our group utilizes ImageJ to determine the image length in pixels. If you opt for the ‘On’ option, ensure that you are using a cropped image of the SEM image (Figure 7). The sample cropped image is included in the Appendix A. Once the scale reading and scale length values are entered, click on the ‘Done’ button. Initiate the analysis by clicking on the ‘Analyze’ button. PoreVision will automatically outline pores based on the predefined settings (Figure 8).

Toggle between the original image and the analyzed image by pressing the ‘View’ button. If the outlined contours are not satisfactory, you can adjust the settings under ‘Analysis Settings’ to enable PoreVision to select pores with greater accuracy (Figure 9).

To remove individual pores, select the desired pore by clicking inside it and then press the ‘Remove’ button. For a clearer view of the image, toggle back to the original image. Any outlines that appear to be erroneous should be removed to ensure a more accurate analysis (Figure 10).

To save pores that you want to keep, select them and then save them by pressing the ‘Save’ button. For visually easier processing, consider turning off ‘Saved Contours’ to minimize the number of outlines displayed on the screen (Figure 11).

Continue the process of removing and saving pores until only the accurately outlined ones identified by PoreVision are saved (Figure 12). This approach guarantees the accuracy of the software’s final analysis.

Once you are satisfied with the outlined pores, save the image by clicking on ‘Save PNG’ and export the data by selecting ‘Save CSV’ (Figure 13).

Interpreting the statistics derived from the analysis of a sample porous gel image provides valuable insights into the characteristics of the pores within the scaffold. For example, in this sample workflow, the minimum pore area observed is 247.37 µm^2^, indicating the presence of relatively small pores within the structure. Conversely, the maximum pore area of 5497.21 µm^2^ suggests the existence of larger pores, potentially contributing to greater interconnectivity within the scaffold. The mean pore area, calculated at 1362.36 µm^2^, serves as a representative measure of pore size distribution within the image. The median pore area of 1024.82 µm^2^ further supports this distribution, indicating a skew towards smaller pores. The standard deviation of pore areas, at 1077.69 µm^2^, underscores the variability in pore size within the scaffold. Moreover, the mean perimeter of 174.30 µm provides additional information regarding the shape complexity of the pores. The mean IPR of 0.55 suggests an average of moderately elongated pore structure within the scaffold. Additionally, the mean pore density of 166.20 pores/mm^2^ illustrates the spatial distribution of pores within the scaffold. These statistics collectively provide a comprehensive understanding of the morphological characteristics of the pores within the analyzed image, laying the groundwork for further interpretation and optimization of scaffold design for tissue engineering applications.

#### 2.2.2. Additional Workflow Techniques

To aid in selecting the ideal analysis settings, the user can click the ‘Original’ button, which will then list a dropdown menu with the options “CLAHE”, “Denoised”, and “Threshold”. Selecting one of these options and then clicking ‘View’ will allow the user to see the effects of changing the respective analysis setting. See Section 4 for example images.

Turning ‘Saved Contours’ to ‘on’ will display contours that were removed by PoreVision. This option displays contours that were removed due to size in blue (affected by maximum and minimum pore size), removed due to hierarchy in magenta (contours that appear to go outside the image border), or removed by the user in orange.

PoreVision also allows users to select a range of sequentially numbered pores to either save or remove more than one contour at a time. Once the user clicks on any contour, two ‘Range’ buttons appear next to the ‘Remove’ and ‘Save’ buttons, which, respectively, allow the user to remove or save a range of contours.

One of the drawbacks of PoreVision is that a particular set of analysis settings may outline some pores very well and other pores poorly. Therefore, the user can save pores and subsequently rerun the analysis using new settings. Each run of the program (clicking ‘Analyze’) is referred to as a “pass”. Pores saved in a particular pass will be labeled with that pass number in the exported CSV file. This allows the user to tune the settings for each pore or a group of pores independent of other pores. It should be emphasized that using different passes for pore detection is a time-consuming process, and the benefit of precision should be weighed against the time cost.

#### 2.2.3. Individual Pore Comparison to Manual ImageJ Measurement

To evaluate the accuracy of PoreVision, program measurements were compared against freehand measurements conducted in ImageJ. The percent deviation for various parameters including area, perimeter, and orientation was utilized for this assessment (Table 3). Percent deviation was calculated using the following formula:(1)$$Percent\;Deviation=\frac{\left|ImageJ\;parameter-PoreVision\;parameter\right|}{ImageJ\;parameter}\times 100\%$$

### 2.3. PoreVision Versus Manual ImageJ Measurements

PoreVision underwent a comparative analysis against the ImageJ protocol used within our research group. Five researchers, all of whom had never used ImageJ or PoreVision before, were tasked with following the ImageJ protocol outlined in Section 2.3 to measure the pores in a designated image. The same researchers analyzed the identical image using PoreVision (see Appendix A). They were also instructed to record the approximate time taken to complete each analysis technique. The average and standard deviation of the data obtained from ImageJ were then compared to those from PoreVision (Table 4). Visual distribution of data can be seen in Figure 14. Detailed individual results can be referenced in the Appendix A.

We found that users who used PoreVision had lower variability in their measurements for area and orientation between each other than when using ImageJ. The variability in perimeter values was roughly the same between PoreVision and ImageJ. PoreVision produced higher variability in averages of measurements when compared to ImageJ, which may be due to the higher sample size that PoreVision outputs and the greater range of smaller to larger pores that may be missed when simply physically scanning the image and manually circling pores. Further, the average time to complete the measurements was equal between the two programs, although PoreVision had a greater variability in the time researchers spent using the program. This may be due to the learning curve when first learning to use PoreVision; though the controls are intuitive, it takes longer to understand how to efficiently use the controls PoreVision provides. The researchers who tested PoreVision were using it for the first time. It is worth noting that lab members already familiar with PoreVision demonstrated an average analysis time of 20 min with consistent results. However, these data were not included as the focus was on evaluating adaptability and performance among new users. Importantly, while PoreVision may not be faster for new users, it allows for the analysis of a higher volume of data with reduced variability between users. Experienced users of both ImageJ and PoreVision may achieve comparable results with smaller datasets; however, PoreVision offers the advantage of handling larger volumes of data with reduced variability.

The comparison of freehand ImageJ measurements to those obtained through PoreVision highlights the high fidelity of PoreVision’s contour detection algorithm. This validation underscores the reliability and accuracy of PoreVision in accurately delineating pores within SEM images, demonstrating its utility as a robust analytical tool in tissue engineering research. Moreover, while this paper primarily focuses on the application of PoreVision in analyzing cryogels, it is important to note that additional testing in our group has demonstrated that PoreVision is capable of detecting pores in other types of porous and fibrous scaffolds (e.g., hydrogels and electrospun scaffolds), though a comparative study with ImageJ for these scaffold types has not yet been conducted. Images of these other scaffold analyses are available in the Appendix A. The versatility of PoreVision extends beyond cryogels, making it a valuable tool for researchers working across various scaffold materials and structures.

### 2.4. PoreVision Limitations and Future Directions

Two-dimensional image-based analyses, such as ImageJ and PoreVision, are used to measure pore sizes due to their ease of use and flexibility in analyzing 2D images obtained from commonly available microscopy. However, 2D image-based analyses, including those performed with both ImageJ and PoreVision, are inherently limited in representing the complexity of 3D pore structures [26,52]. Measurements derived from 2D slices may fail to capture the full interconnectivity and depth of pores, potentially leading to sampling bias and underestimating pore sizes. In contrast, 3D techniques, such as Brunauer–Emmett–Teller (BET) analysis or microCT, offer a more comprehensive characterization by providing quantitative data on surface area, pore volume, and connectivity throughout the entire structure [40,53]. While these 3D methods are more accurate, they require specialized equipment and are costlier, limiting their widespread use. PoreVision development emphasizes simplifying 2D analyses for rapid scaffold characterization while acknowledging the inherent limitations of 2D approaches.

Another concern revolves around the accuracy of the analysis. While PoreVision offers users the flexibility to adjust settings and conduct multiple passes to improve accuracy, the onus still falls on the user to exercise caution. It is not a one-click solution, necessitating careful attention and validation of results. Future direction includes the development of compatibility with Mac OS, broadening its accessibility to a wider user base. Additionally, integrating machine learning techniques could enhance pore identification capabilities, offering a more automated and refined analysis process.

## 3. Conclusions

PoreVision presents an advancement in tissue engineering research, offering a user-friendly, and consistent approach to characterizing porous gels from SEM images. Through its efficient pore characterization capabilities, including shape and orientation analysis, PoreVision facilitates precise and high-throughput analysis, which can accelerate analysis in scaffold research. Comparative analysis against traditional methods demonstrates PoreVision’s efficiency and measurement precision. Overall, PoreVision stands as a promising tool for researchers, enabling robust pore characterization and advancing scaffold design for regenerative medicine applications.

## 4. Materials and Methods

### 4.1. Porous Gel Scaffold Fabrication

Chitosan-gelatin cryogels were used to aid the development of this software. Cryogels were fabricated following established procedures [48]. Initially, 3 mL syringes (Fisher Scientific, Fair Lawn, NJ, USA) were pre-frozen at −23 °C for 6 h. A 10 mL aliquot of 1% acetic acid (Fisher Scientific, Fair Lawn, NJ, USA) in deionized water (DI) was prepared and divided into 8 mL and 2 mL portions, each transferred into separate scintillation vials. Subsequently, low-viscosity chitosan (80 mg, Mw = 1526.464 g/mol; MP Biomedicals, Solon, OH, USA) was added to the 8 mL portion and vortexed for 30 s before being placed on a mechanical spinner for 1 h. Gelatin from cold water fish skin (320 mg, Mw = 60 kDa; Sigma-Aldrich, St. Louis, MO, USA), was then introduced to the 8 mL aliquot and subjected to mechanical spinning for an additional hour to ensure complete dissolution. The remaining 2 mL aliquot of 1% acetic acid was combined with glutaraldehyde (50% aq. Soln, Thermo Scientific Chemicals, Waltham, MA, USA) to produce a 1% glutaraldehyde solution. Following this, both the 8- and 2 mL scintillation vials were refrigerated at 4 °C for 1 h. After the cooling period, the solutions from the 8- and 2 mL vials were mixed by decanting between the vials and promptly dispensed into the pre-frozen syringes. The filled syringes were then immediately transferred to a −23 °C freezer and allowed to crosslink at subzero temperatures for 18 h, followed by thawing at room temperature.

### 4.2. Scanning Electron Microscopy (SEM)

SEM imaging of the cryogel pore structure was conducted using a VEGA3 SEM instrument (TESCAN, Brno, Czech Republic). The cryogels were initially frozen at −80 °C for 1 h before undergoing lyophilization overnight using a FreeZone Freeze Dryer (Labconco, Kansas City, MO, USA). The samples were cut transversely such that an interior cross section may be analyzed within each sample. Subsequently, the samples were mounted on an aluminum stub and sputter coated with gold using a HUMMER 6.2 sputter coater (Anatech, Sparks, NV, USA) for 240 s at 15 mA under the pulse setting to prevent overheating. Images were taken at approximately 200× magnification for analysis.

### 4.3. Manual Pore Measurements

#### 4.3.1. ImageJ Pore Analysis

ImageJ software (version 1.53t) was used to analyze the pore area in cryogel scaffolds. The line function within ImageJ was used to establish the scale of the SEM image using the provided scale bar, with units set to micrometers for accurate measurements. The elliptical selections tool was used to measure the area of a representative pore by outlining the approximate perimeter of the pore. This process was repeated 60 times, with 15 measurements taken from each quadrant of the image to ensure representative sampling as shown in Figure 15. The recorded data were then saved in an Excel file, with the area values (µm^2^), Skewness, Perimeter (µm), and Feret’s diameter (µm) measurements utilized for subsequent statistical analysis. The full protocol is added in Appendix A.

#### 4.3.2. Individual Pore Comparison to Manual ImageJ Measurements

Ten numbers, from 1 to 211, were randomly generated using Google Sheets. The ten respectively numbered pores were selected from a completed PoreVision analysis and found in the blank SEM image that was imported into ImageJ (Figure 16). The line function within ImageJ was utilized to establish the scale of the SEM image using the provided scale bar, with units set to micrometers. Using the author’s best judgment, the freehand selections tool was used to outline the perimeter of the identified pores, without referencing the outlines in the completed PoreVision analysis. The recorded data were then saved in an Excel file, with the area values (µm^2^), Perimeter (µm), and orientation (°) measurements utilized for subsequent statistical analysis.

### 4.4. Software

#### 4.4.1. Requirements

PoreVision is intended to serve as a tool to quickly and repeatably characterize SEM images of porous gels, particularly cryogels. It offers a suite of analysis tools within an intuitive graphical user interface.

PoreVision requires Windows and a screen resolution less than or equal to 1440p since 4k may cause click tracking/DPI scaling issues in its current version.

The image segmentation performed by PoreVision to detect pores is based on common computer vision techniques. The Python library OpenCV is used for many operations. PoreVision also makes use of several other commonly used data analysis libraries, such as NumPy and Pandas, that many researchers who use Python will likely already be familiar with. What follows is an overview of the steps that PoreVision follows to analyze an image.

#### 4.4.2. Content Localized Adaptive Histogram Equalization (CLAHE)

First, CLAHE is performed on the image by OpenCV’s “createCLAHE” and “clahe.apply” functions. This algorithm breaks an image into kernels of a user-specified size and applies a histogram equalization to their grayscale values, with a contrast cutoff to ensure that contrast is not boosted too high in localized areas of the image. In effect, this process corrects light and dark regions of the image so that further analysis is equally applied to the whole image (Figure 17). Lighter regions are darkened, and darker regions are lightened. Since darkness corresponds to depth in an SEM image, if the cross section being imaged is not level, some regions will appear darker than others, and pore detection will not be consistently applied.

#### 4.4.3. Denoising

Next, the image is denoised according to the Fast Non-Local Means Denoising algorithm described by Buades et al., as implemented by OpenCV in their “fastNlMeansDenoising” function [49]. This is performed to remove stray pixels and noise that can be erroneously identified as contours, such as cracks and folds in a scaffold. The size of this kernel, as well as the strength of the operation, can be specified by the user. In many cases, lowering/raising denoising size or denoising strength will have a similar effect (Figure 18). Note that this operation smooths the edges of contours, which can drastically affect the isoperimetric ratio (IPR) of the pores without changing their area.

#### 4.4.4. Cutoff

Next, the image undergoes binary thresholding according to a user-defined cutoff value. Any pixel with a grayscale value under the cutoff threshold is turned black, and any pixel over the threshold is turned white. This creates a binary black-and-white image. Raising the cutoff value means that lighter regions are now turned black and rendered eligible for contour detection, which often effectively “erodes” the edges of the contours (Figure 19).

#### 4.4.5. Contour Detection

Contour detection, the most critical step, is performed next. PoreVision uses the OpenCV “findContours” function, an application of the algorithm described by Suzuki and Abe, which iterates through the image and traces the delineations between black and white pixels, returning a list of contours [50]. These contours are lists of adjacent pixels, (x, y) coordinate pairs, that correspond to the edges of the black regions of the thresholded image, which correspond to the edges of the darkest regions (pores) of the original image. Each distinct “loop” of pixels is a separate contour, corresponding to a separate pore. Detected contours are then either accepted or rejected according to the user-specified size values. All contours bordering the edge of the image, or inside another contour, are automatically rejected.

Once a list of accepted contours has been created, the program performs several image analysis techniques on the entire list. Areas and perimeters are measured directly, and IPRs are calculated from them. Orientation vectors are found by fitting an ellipse to the contour according to the algorithm described by Fitzgibbon and Fisher using OpenCV’s “fitEllipse” function, and then subtracting the magnitude of the minor axis from that of the major axis of the fitted ellipse, yielding a vector that relates both the direction and magnitude of the orientation of the pore on an XY plane [51]. Note that pores do not have inherent directionality, so all pores are assumed to be pointing right. Location is found by calculating the average XY coordinates of all pixels contained within the contour.

Once this analysis is complete, the resulting contours are drawn onto the original image, along with any optional visual aids as specified by the user.

#### 4.4.6. Program Fidelity

Like any computer vision application, PoreVision will occasionally misread a feature in the provided image. For this reason, PoreVision allows the user to selectively remove contours that the program erroneously identified. PoreVision also allows the user to save contours between multiple passes if all the appropriate contours are not being detected by a single pass. In this context, a “pass” refers to an individual run of the program over the image; a single pass of the program occurs every time the ‘Analyze’ button is pressed.

It is important to note that it is still up to the discretion of the user to gauge whether PoreVision is correctly identifying pores or not based on the supplied parameters and then adjust those parameters if necessary. Once satisfactory parameters are identified, the same analysis can be applied for any number of images in a dataset, assuming the imaging settings were also held constant. The predefined parameters may not be suitable for all applications.

#### 4.4.7. Saved Data

PoreVision allows a user to save the processed images, as well as export a comma-separated values (CSV) file that contains data pertaining to each individual pore. This CSV file contains the number, area, perimeter, isoperimetric ratio, orientation vectors, center coordinates, minimum and maximum feret diameters of the fitted ellipses, and save profile of each pore. If contours are saved through multiple passes, PoreVision will track which settings were used for each contour and assign each one a “Save Profile”, which consists of all analysis settings for that pass, also listed in the CSV file.

### 4.5. Statistics

Data collection and basic analysis were conducted using Microsoft Excel and Google Sheets. Statistical analyses were then performed with GraphPad Prism, using a significance level of 0.05. F tests were conducted to assess the variances between groups. If the F test indicated that the variances were not different, unpaired t-tests were used to determine significant differences between the groups. However, if the F-test suggested unequal variances, unpaired t-tests with Welch’s correction were applied.

## Figures and Tables

**Figure 1 gels-11-00132-f001:**
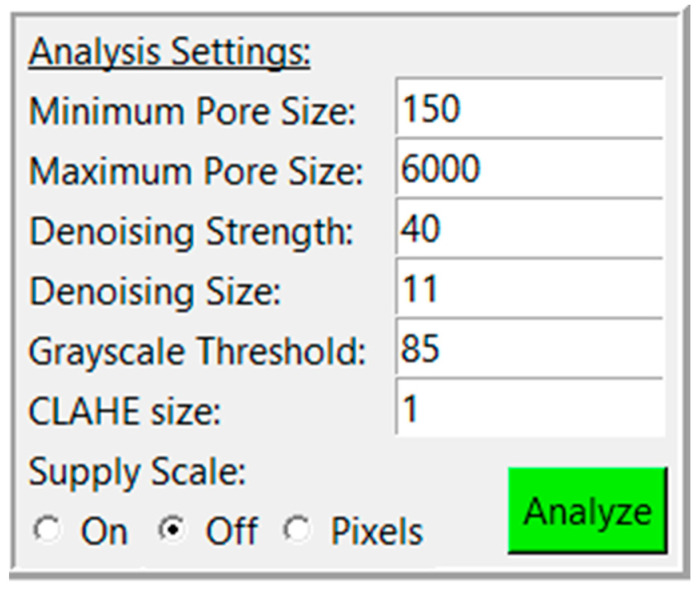
Analysis settings with adjustable settings and analyze button.

**Figure 2 gels-11-00132-f002:**
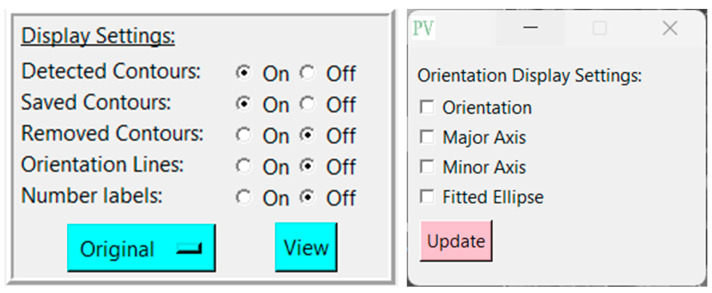
Display settings and orientation display settings with adjustable options.

**Figure 3 gels-11-00132-f003:**
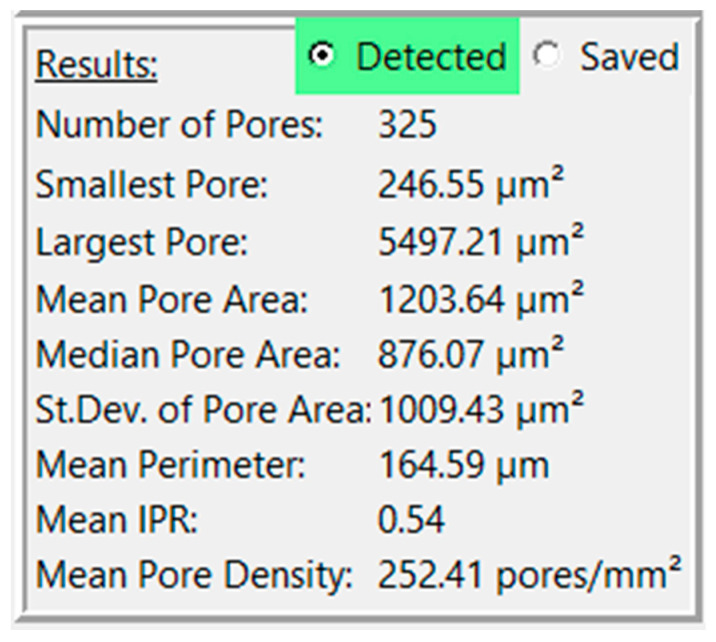
View of results for a sample image.

**Figure 4 gels-11-00132-f004:**
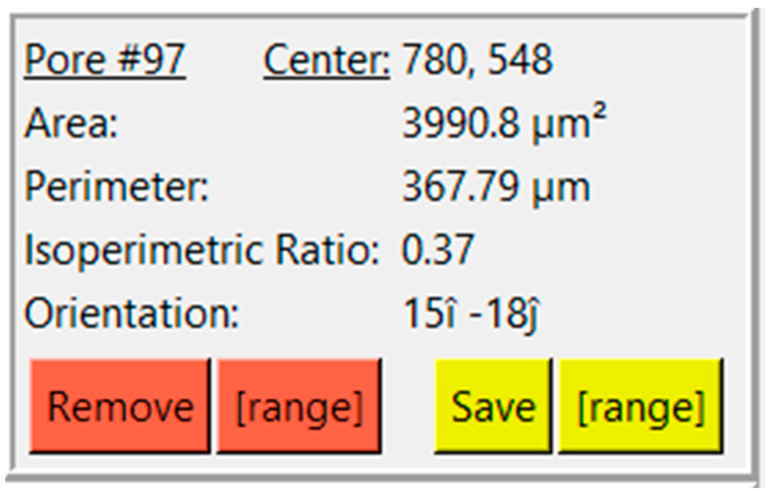
View of results and options for a selected pore.

**Figure 5 gels-11-00132-f005:**
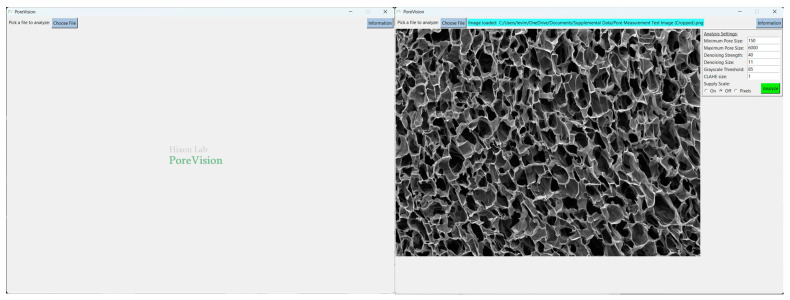
PoreVision start screen and screen after selecting an image file.

**Figure 6 gels-11-00132-f006:**
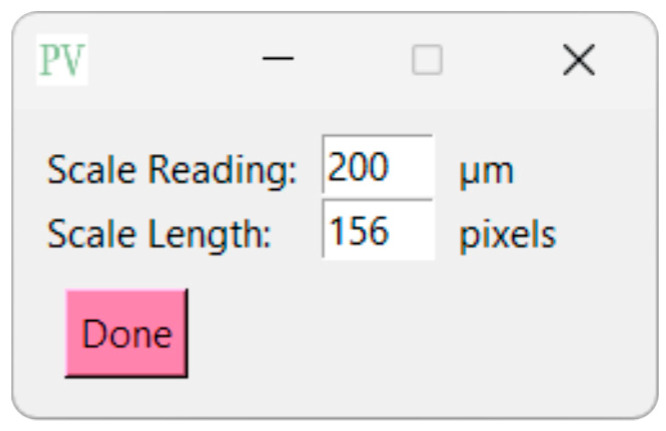
View of supply scale window.

**Figure 7 gels-11-00132-f007:**
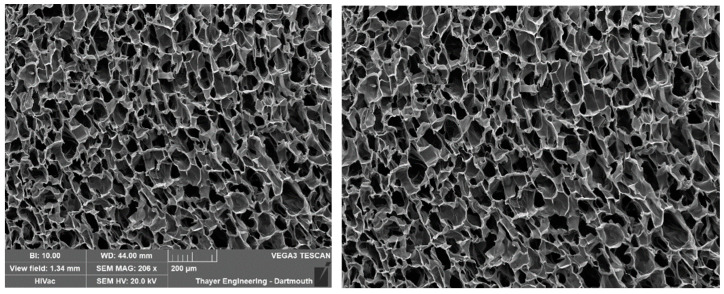
Original SEM image and cropped SEM image.

**Figure 8 gels-11-00132-f008:**
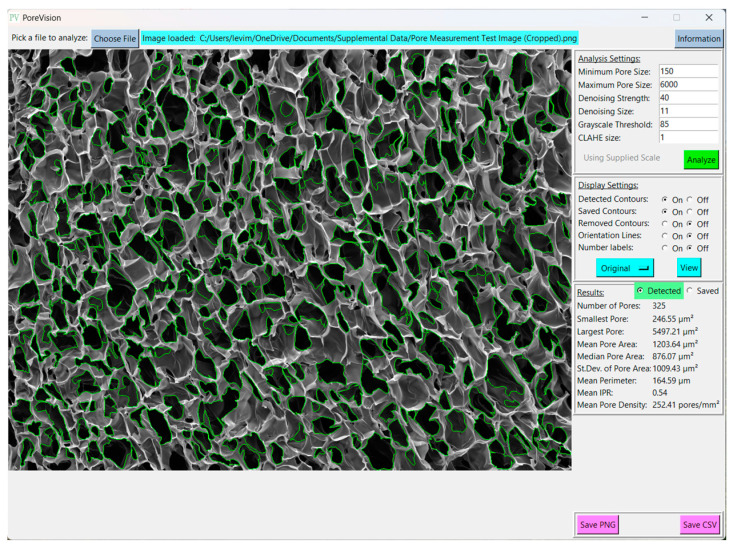
Initial analysis results with predefined settings.

**Figure 9 gels-11-00132-f009:**
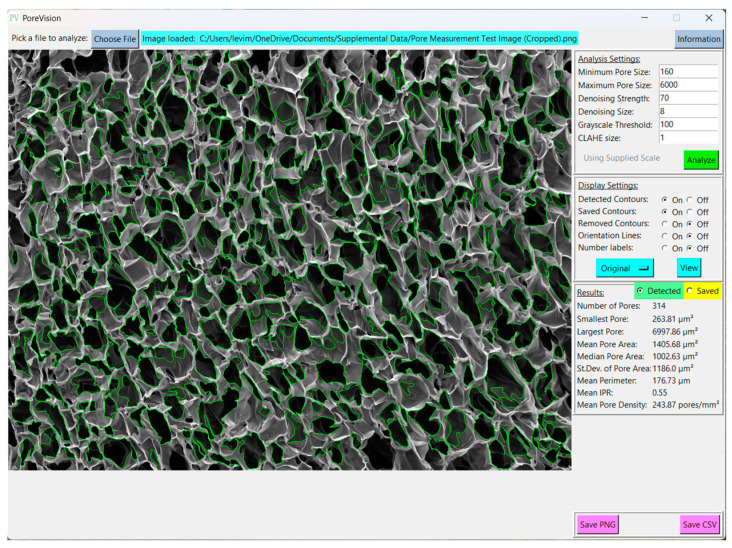
View of results with adjusted analysis settings.

**Figure 10 gels-11-00132-f010:**
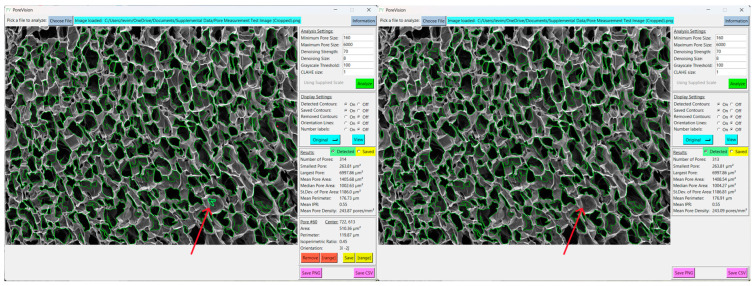
Erroneous pore (red arrow) before and after removing. It was removed because it is a shadow that was outlined inside of a larger pore.

**Figure 11 gels-11-00132-f011:**
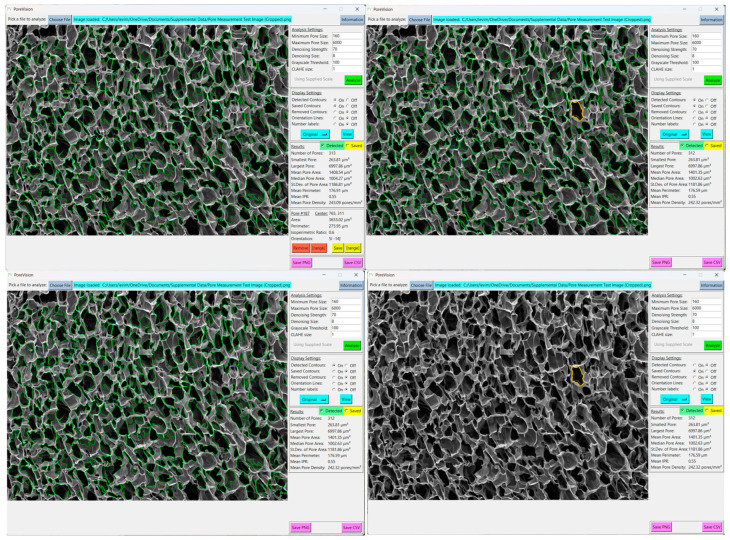
Views of a pore selected (top left), a pore saved (top right), analysis with saved pore contours off (bottom left), analysis with saved contours only (bottom right).

**Figure 12 gels-11-00132-f012:**
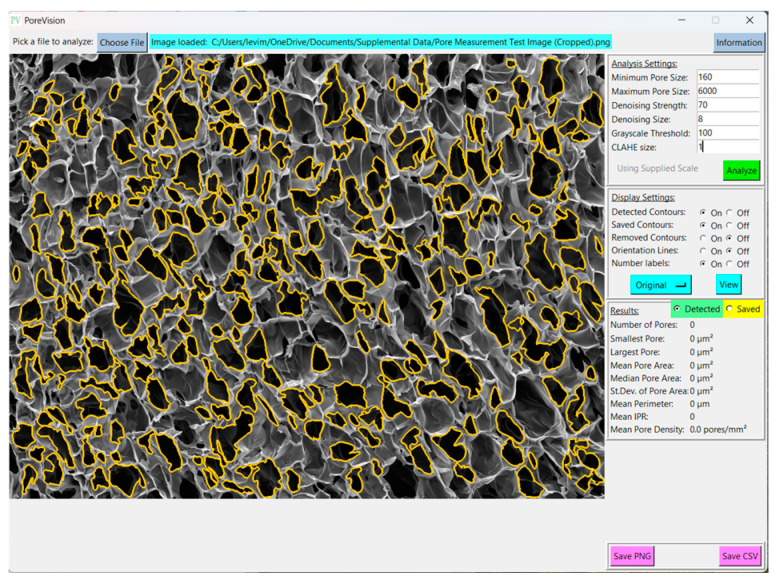
View of fully processed image.

**Figure 13 gels-11-00132-f013:**
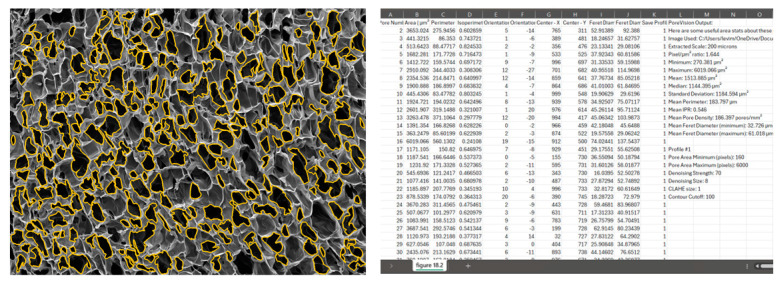
View of exported PNG and CSV files.

**Figure 14 gels-11-00132-f014:**
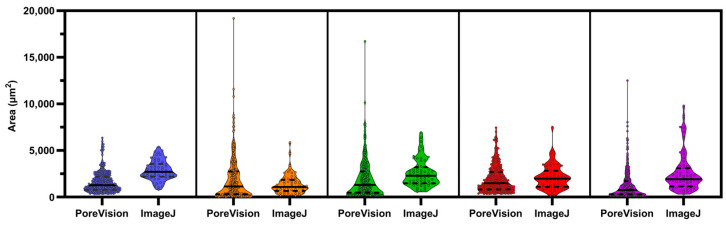
Violin plot with individual data points of PoreVision versus manual ImageJ measurements from researchers 1 through 5. Solid lines are medians; Dashed lines are quartiles.

**Figure 15 gels-11-00132-f015:**
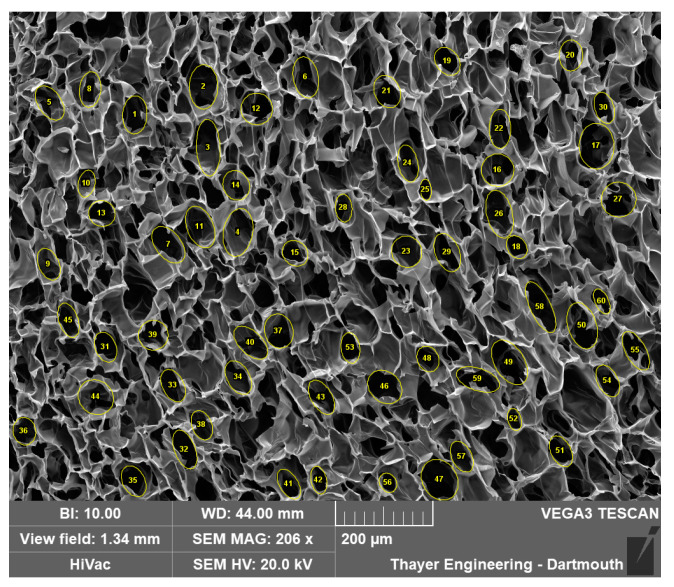
Example manual analysis using ImageJ.

**Figure 16 gels-11-00132-f016:**
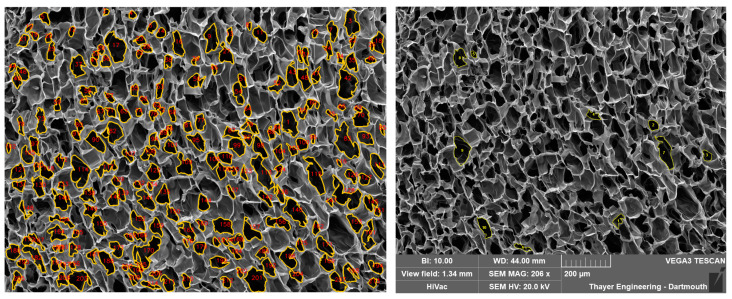
Detected pores in PoreVision vs. Freehand selections in ImageJ.

**Figure 17 gels-11-00132-f017:**
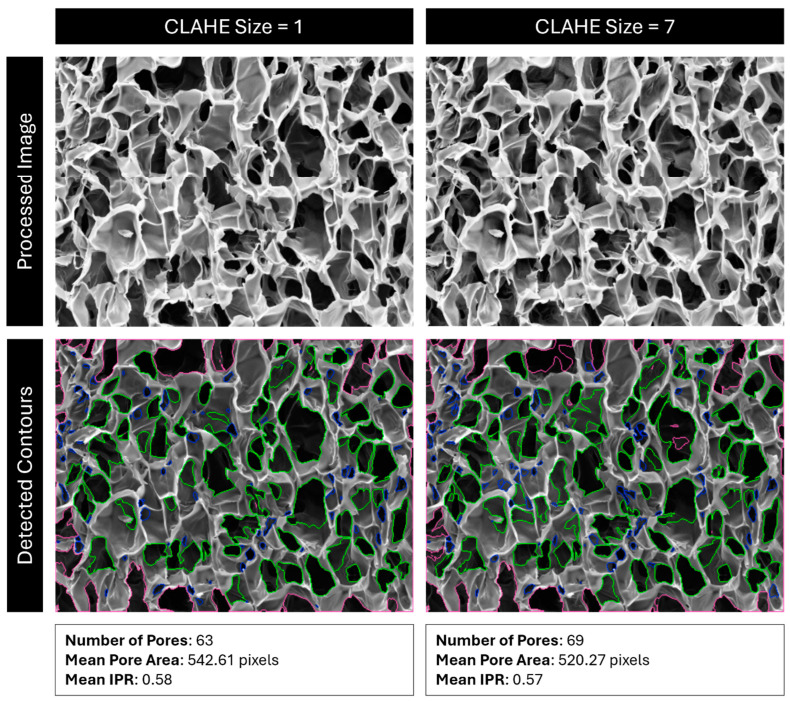
Comparison of detected pores using CLAHE size 1 vs. 7. Red outlines are pores outside the analysis boundary, blue outlines are pores removed for being too small (most likely dust, cracks, or folds), and green outlines are identified pores.

**Figure 18 gels-11-00132-f018:**
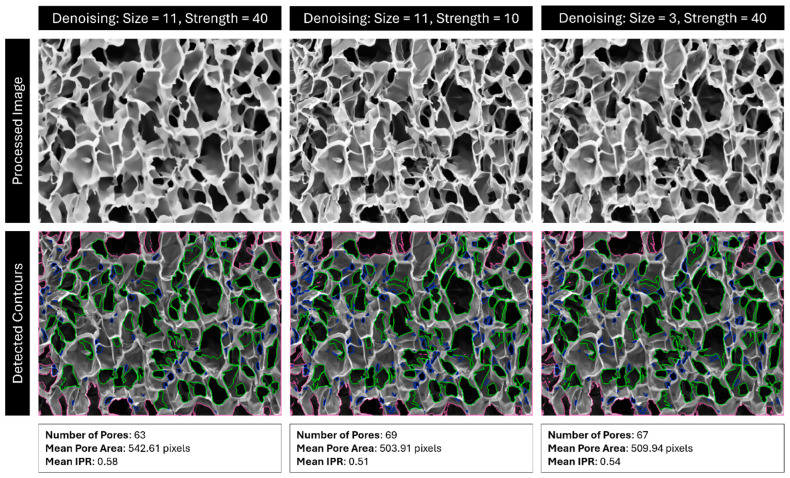
Comparison of detected pores using denoising with size 11 and strength 40, size 11 and strength 10, and size 3 and strength 40. Red outlines are pores outside the analysis boundary, blue outlines are pores removed for being too small (most likely dust, cracks, or folds), and green outlines are identified pores.

**Figure 19 gels-11-00132-f019:**
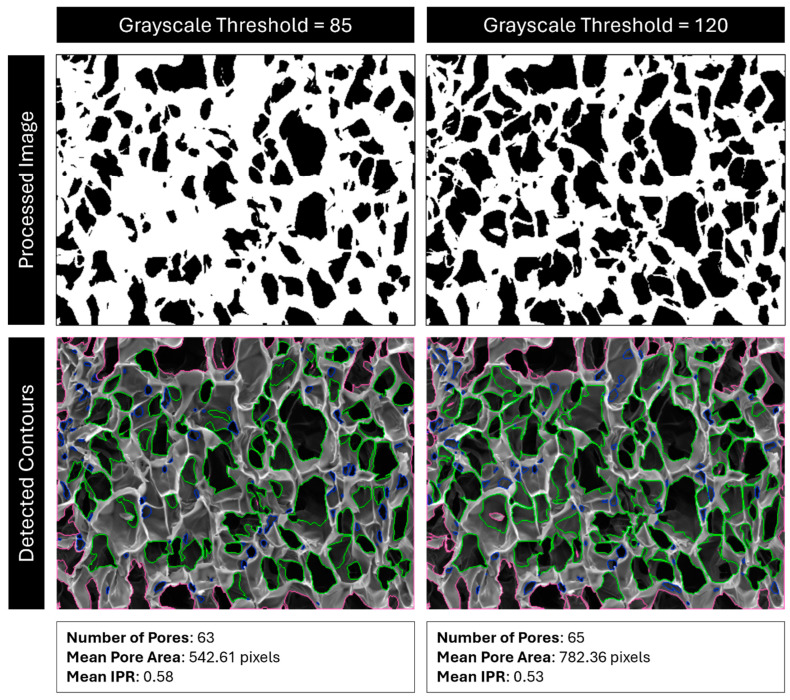
Comparison of detected pores with cutoff values of 85 vs. 120. Red outlines are pores outside the analysis boundary, blue outlines are pores removed for being too small (most likely dust, cracks, or folds), and green outlines are identified pores.

**Table 1 gels-11-00132-t001:** Description of analysis settings. See Section 2.4 for explanations of the algorithms for each setting. See PoreVision Protocol in Appendix A for a user friendly guide on how to select and adjust each parameter.

Setting	Description
Minimum Pore Size:	The minimum size of pore, in pixels, that the program will consider for analysis. Can be useful for preventing the reading of noise and other features such as dust, cracks, and folds in the cryogel that are sometimes read as contours (i.e., detected pores that are too small may not be real pores)
Maximum Pore Size:	The maximum size of contour, in pixels, that the program will consider for analysis
Denoising Strength:	The strength of the denoising operation
Denoising Size:	The size of the kernel (in pixels) used for the denoising operation
Grayscale Threshold:	The cutoff value used for binary thresholding
CLAHE size:	The size of the kernel used for CLAHE
Supply Scale:	Allows the user to analyze the image in terms of pixels, a supplied scale, or attempt to read the scale from the image (some models of Tescan SEMs work)

**Table 2 gels-11-00132-t002:** Descriptions of each display setting.

Setting	Description
*Detected Contours:*	This option displays the contours detected by the last analysis pass in green on the original image. It is on by default.
*Saved Contours:*	This option displays contours saved by the user in gold. It is on by default. These contours will persist between analysis passes.
*Removed Contours:*	This option displays contours removed due to size (blue), hierarchy (magenta), or by the user (orange).
*Orientation Lines:*	This option displays different options to visualize the “directionality” of pores including orientation, major axis, minor axis, and fitted ellipse.
*Number Labels:*	This option prints a number in red on each detected pore on the resulting image.

**Table 3 gels-11-00132-t003:** Percent deviations between freehand ImageJ and PoreVision.

	PoreVision			ImageJ			% Difference		
	Area (µm^2^)	Perimeter (µm^2^)	Orientation (°)	Area (µm^2^)	Perimeter (µm^2^)	Orientation (°)	Area (µm^2^)	Perimeter (µm^2^)	Orientation (°)
1	1000	151	51	1271	192	78	21.3%	21.3%	34.0%
2	1262	139	124	1269	134	137	0.5%	3.4%	9.6%
3	1119	137	108	1123	137	129	0.4%	0.4%	16.3%
4	1238	191	170	1096	166	140	13.0%	14.9%	21.0%
5	487	88	121	442	84	129	10.2%	4.4%	6.5%
6	866	179	163	876	178	163	1.1%	0.4%	0.2%
7	4730	421	111	4772	393	120	0.9%	7.2%	8.0%
8	2710	275	96	2696	259	80	0.5%	6.1%	19.7%
9	4942	362	101	5228	312	100	5.5%	16.0%	1.0%
10	2613	250	115	2551	242	125	2.4%	3.2%	7.7%
					**Average % Difference:**		**5.6%**	**7.7%**	**12.4%**

**Table 4 gels-11-00132-t004:** Results from ImageJ analysis vs. PoreVision analysis of the same image. * Indicates a significant difference between groups (*p* < 0.05).

		Mean of Averages	Standard Deviation of Averages	Mean of Standard Deviations
ImageJ	**Area (µm** ** ^2^ ** **)**	2313.2	581.6	1399.4
	**Perimeter (µm)**	172.6	24.0	52.0
	**Orientation (°)**	111.8	5.2	24.0
	**Time (min)**	30.0	6.1	6.1
PoreVision	**Area (µm** ** ^2^ ** **)**	1728.8	302.6	1652.2
	**Perimeter (µm)**	214.0	24.3	139.4
	**Orientation (°)**	101.0	1.6	37.2
	**Time (min)**	31.0	13.4	13.4
% Difference	**Area (µm** ** ^2^ ** **)**	−25.3%	−48.0%	18.1%
	**Perimeter (µm)**	* 24.0%	1.2%	* 168.3%
	**Orientation (°)**	* −9.7%	* −68.8%	* 55.0%
	**Time (min)**	3.3%	119.1%	119.1%

## Data Availability

The original contributions presented in this study are included in the article/Appendix A. Further inquiries can be directed to the corresponding author.

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
