# Peer review of "PoreVision: A Program for Enhancing Efficiency and Accuracy in SEM Pore Analyses of Gels and Other Porous Materials"

_gels, 2025, doi:10.3390/gels11020132_

Round 1

Reviewer 1 Report

Comments and Suggestions for Authors

This well-organized paper presents an effective method for enhancing pore measurement using modified free software. However, there are some suggestions for revision before it can be acceptable.

Comments:

Abstract: I recommend that the authors include some simulation results to support and validate their research; therefore, this section needs to be revised.

Introduction: 

Lines 59-61: Since ImageJ is a widely used analysis software, the authors should expand this section.

Methods: 

Page 4, Lines 144-145: Could the authors provide a link or reference to PoreVision, which claims to be available for free download? I was unable to locate it through a Google search.

Results: 

Lines 246-247: I cannot find the mentioned zip file in the supplementary information; please confirm its availability. Thank you.

Figure 8: Please clarify the pore data where the standard deviation of the pore area is greater than both the mean and median pore area, and note that the mean and median pore diameters are not provided.

Figure 10: Please rearrange the presentation of this image and improve its resolution.

Author Response

Dear Editors and Reviewers,

We are pleased to submit the revised version of gels-3363178: “PoreVision: A Program for Enhancing Efficiency and Accuracy in SEM Pore Analyses of Gels and Other Porous Materials.” We appreciate the reviewers' valuable feedback and have thoroughly addressed each of their concerns as detailed below. All revisions to the manuscript are indicated as tracked changes. Additionally, we have included a clean version of the manuscript without the tracked changes. We enhanced both the text and figures, and added additional information to enrich the paper; these modifications have notably enhanced the paper and improved its clarity.

1. Abstract: I recommend that the authors include some simulation results to support and validate their research; therefore, this section needs to be revised.

  • Response: We have added text in the abstract that states simulation results.

2. Introduction: Lines 59-61: Since ImageJ is a widely used analysis software, the authors should expand this section.

  • Response: Additional elaboration was included discussing methods that employ ImageJ for pore measurements.

3. Methods: Page 4, Lines 144-145: Could the authors provide a link or reference to PoreVision, which claims to be available for free download? I was unable to locate it through a Google search.

  • Response: PoreVision will be available on our public lab group’s website after the manuscript’s acceptance. We did not want to publish unreviewed instructions or data before the peer review process was complete.

4. Results: Lines 246-247: I cannot find the mentioned zip file in the supplementary information; please confirm its availability. Thank you.

  • Response: We have added the folder “PoreVision Program” to the supplemental data zip file. It seems the folder was accidentally left out during the manuscript upload process.

5. Figure 8: Please clarify the pore data where the standard deviation of the pore area is greater than both the mean and median pore area, and note that the mean and median pore diameters are not provided.

  • Response: In figure 8, the standard deviation (1009.43 μm2) is less than the mean (1203.64 μm2) but greater than the median (164.59 μm2) pore area. These values are a result of the asymmetry of the data points caused by a large right skew; most of the pores are small and only a few are large. Physically this makes sense within the context of cryogels because of their tendency to form smaller pores (smaller ice formations) than it is to make large pores (larger ice formations).
    • Lines 407-419 discuss an example of interpreting the output of the particular analysis demonstrated in Figure 8.
    • Added language noting the data that is not provided in the user interface.

6. Figure 10: Please rearrange the presentation of this image and improve its resolution.

  • Response: We have improved the resolution of images. Please note that image resolution is reduced when converting from the word document to the PDF document (which MDPI does for review). Gels will be able to include high resolution images in the final manuscript from the images we have attached and submitted separately.

Reviewer 2 Report

Comments and Suggestions for Authors

Oleversky et al. developed user-friendly software to measure pore sizes in SEM images. The work is well-written, easy to follow, relevant to the journal´s target audience, and within the scope of this one. At this point, I have a few comments for the authors (please see below); therefore, I recommend a minor revision.

1)      Were the SEM images considered in this study obtained from a superficial view, or do they correspond to cross-sectional views of the cryogels? Please clarify this point in the methodology section.

2)      Table 1 is apparently not referenced within the text.

3)      In Table 1, the two superior rows (Minimum and maximum pore sizes), I recommend that the authors give some advice to potential users about how to select these parameters, as they will depend on the use of other tools.

4)      Why are the “too small” pores discarded? Please discuss this within the text. It should be considered within the program's limitations.

5)      A relevant point is that the authors only employed cryogels, yet they mentioned that other types of hydrogels were also measured (data not shown). Why? Those results should be provided, and even if they do not match what is expected, they should provide information about the software's limitations. At this point, the conclusion regarding the type of samples able to be measured is not supported.

6)      It can be interesting to compare a pore size distribution plot employing ImageJ vs PoreVision.

7)      Understanding that ImageJ is highly employed to measure image pore sizes and that the authors offer some discussion about the limitations of this type of 2D approach, the discussion section is still too poor regarding this point. Why is ImageJ or any visualization software employed to measure pores? Are those measured pores representative of a sample? How are the obtained results compared to a 3D measurement (e.g., employing BET)?  

8)      The conclusion regarding time optimization is not supported. From the data provided by the authors, the time employed for the new users is approximately similar. Then, the authors mention that trained people can perform the analysis in 20 minutes on average using PoreVision; however, I am guessing that the same people with the right training also decrease the time employed using ImageJ. I think this must be corrected within the text and conclusion section. It is not bad if it is not quicker; what I think is relevant is that it allows us to consider a higher volume of data with less (apparent) variability. 

Author Response

Dear Editors and Reviewers,

We are pleased to submit the revised version of gels-3363178: “PoreVision: A Program for Enhancing Efficiency and Accuracy in SEM Pore Analyses of Gels and Other Porous Materials.” We appreciate the reviewers' valuable feedback and have thoroughly addressed each of their concerns as detailed below. All revisions to the manuscript are indicated as tracked changes. Additionally, we have included a clean version of the manuscript without the tracked changes. We enhanced both the text and figures, and added additional information to enrich the paper; these modifications have notably enhanced the paper and improved its clarity.

1. Were the SEM images considered in this study obtained from a superficial view, or do they correspond to cross-sectional views of the cryogels? Please clarify this point in the methodology section.

  • Response: We have added clarification that cross sections of the cryogels were used in the study.

2. Table 1 is apparently not referenced within the text.g

  • Response: We have added a reference to table 1 in the text.

3. In Table 1, the two superior rows (Minimum and maximum pore sizes), I recommend that the authors give some advice to potential users about how to select these parameters, as they will depend on the use of other tools.

  • Response: We have added wording that directs users to the supplemental materials where a user-friendly guide is written that provides guidance on how to select and adjust each parameter.

4. Why are the “too small” pores discarded? Please discuss this within the text. It should be considered within the program's limitations.

  • Response: Clarification has been added to figures 3-5 and table 1.

5. A relevant point is that the authors only employed cryogels, yet they mentioned that other types of hydrogels were also measured (data not shown). Why? Those results should be provided, and even if they do not match what is expected, they should provide information about the software's limitations. At this point, the conclusion regarding the type of samples able to be measured is not supported.

  • Response: We have included images with PoreVision analyses of other scaffolds types in the supplemental information. We also amended our wording in the text to clarify that we have only tested whether PoreVision can detect pores in other types of scaffolds; we have not done a ImageJ vs Porevision comparative analysis between these other scaffold types.

6. It can be interesting to compare a pore size distribution plot employing ImageJ vs PoreVision.

  • Response: A distribution graph has been added.

7. Understanding that ImageJ is highly employed to measure image pore sizes and that the authors offer some discussion about the limitations of this type of 2D approach, the discussion section is still too poor regarding this point. Why is ImageJ or any visualization software employed to measure pores? Are those measured pores representative of a sample? How are the obtained results compared to a 3D measurement (e.g., employing BET)?

  • Response: We have added a paragraph in discussion to reflect limitations of 2D approaches as compared to 3D approaches.

8. The conclusion regarding time optimization is not supported. From the data provided by the authors, the time employed for the new users is approximately similar. Then, the authors mention that trained people can perform the analysis in 20 minutes on average using PoreVision; however, I am guessing that the same people with the right training also decrease the time employed using ImageJ. I think this must be corrected within the text and conclusion section. It is not bad if it is not quicker; what I think is relevant is that it allows us to consider a higher volume of data with less (apparent) variability.

  • Response: We have removed unsupported claims and amended text to reflect that PoreVision may produce similar time analysis but can handle a higher volume of data with less variability.

Reviewer 3 Report

Comments and Suggestions for Authors The entire manuscript content is more like an instruction manual for the mature software ImageJ, rather than a scientific research.

Author Response

Dear Editors and Reviewers,

We are pleased to submit the revised version of gels-3363178: “PoreVision: A Program for Enhancing Efficiency and Accuracy in SEM Pore Analyses of Gels and Other Porous Materials.” We appreciate the reviewers' valuable feedback and have thoroughly addressed each of their concerns as detailed below. All revisions to the manuscript are indicated as tracked changes. Additionally, we have included a clean version of the manuscript without the tracked changes. We enhanced both the text and figures, and added additional information to enrich the paper; these modifications have notably enhanced the paper and improved its clarity.

Thank you for your feedback and for reviewing our manuscript. We appreciate your observation regarding the instructional tone of the content. Since our primary aim is to introduce PoreVision as a practical and scientific tool for pore characterization, some instructional details were included intentionally to guide potential users through its features and scientific capabilities. However, we understand the importance of balancing instruction with scientific context. To address this, we have ensured that the scientific contributions—such as the improved reproducibility, handling of larger datasets, and reduced variability in key measurements—are emphasized alongside the practical guidance. This will help clarify that the manuscript demonstrates not only how PoreVision operates but also how it advances research in tissue engineering.

Round 2

Reviewer 3 Report

Comments and Suggestions for Authors

I can accept the revised manuscript.